# Closing the Sim-to-Real Gap in Network Spreading Processes via GPU-Accelerated Distributional RL

**Heman Shakeri** [1]

## Abstract

Controlling spreading processes on networks, such as epidemics, information cascades, and product adoption, requires policies that perform on realistic stochastic dynamics, not just tractable approximations. Yet policies trained on standard simplifications (mean-field ODEs, Markovian dynamics) suffer severe performance degradation at deployment. We trace this sim-to-real gap to three theoretical pathologies: Optimism Bias, where deterministic approximations systematically underestimate variance via Jensen's inequality; Hub Blindness, where global state aggregation obscures the super-spreaders driving scale-free networks; and the Valley of Death, where mean-value critics fail to navigate the bimodal nature (extinction vs. viral) of cascade outcomes. We resolve these challenges through two synergistic contributions. First, the Stratified Mean-Field Observer partitions nodes by influence tier, preserving hub dynamics at $\mathcal{O}(N)$ cost while producing fixed-dimensional observations that enable zero-shot transfer across network scales and topologies. Second, we show that distributional RL via Truncated Quantile Critics improves risk-aware control of bimodal cascades. Trained on a GPU-accelerated simulator supporting non-Markovian renewal dynamics, our approach achieves $59\times$ improvement over Markovian baselines and robust zero-shot transfer to real-world social networks (Facebook, Twitter, YouTube), significantly mitigating the simulation-to-reality gap.

[1]School of Data Science, University of Virginia, Charlottesville, VA, USA. Correspondence to: Heman Shakeri <hs9hd@virginia.edu>.

*Proceedings of the 43$^{rd}$ International Conference on Machine Learning*, Seoul, South Korea. PMLR 306, 2026. Copyright 2026 by the author(s).

## 1. Introduction

Reinforcement Learning has emerged as a dominant paradigm for sequential decision-making, with agents deriving intelligence through interaction rather than explicit programming (Sutton et al., 1998). In domains where real-world experimentation is costly, dangerous, or temporally infeasible—one cannot release a virus to test containment policies or crash markets to evaluate trading strategies—training occurs exclusively in simulation (Salvato et al., 2021). This reliance on synthetic environments has unearthed a formidable barrier: the "Sim-to-Real Gap," the degradation of agent performance when transferred from simulation to deployment (Zhao et al., 2020). While extensively cataloged in robotics, where it manifests as dynamics mismatch (simplified physics), perceptual mismatch (idealized rendering), and actuation latency (Muratore et al., 2022), a parallel and equally critical gap exists in Graph-Based Reinforcement Learning (GRL) (Liu et al., 2025).

In GRL, the "environment" is not Euclidean space governed by Newtonian mechanics but a topological structure governed by spreading dynamics, i.e. how information, viruses, or influence propagate through networks (Meirom et al., 2021). Note that throughout this paper, we use general network-science terminology interchangeably with domain-specific terms (e.g., utility/profit as reward, activation/conversion as the target state transition, and hub retention as the successful protection of high-degree nodes).

The graph sim-to-real gap arises from three distinct pathologies (Almasan et al., 2022). First, *Topological Mismatch*: agents trained on synthetic generators like Barabási–Albert capture power-law degree distributions but miss community structure, clustering coefficients, and assortativity characteristic of real social networks (Broido and Clauset, 2019). Second, *Dynamical Mismatch*: standard Markovian simulators (SIR, Independent Cascade) assume memoryless transitions, but real-world influence is non-Markovian, i.e. the probability of adoption depends on interaction history, not just current state (Min and San Miguel, 2018). Third, *Observational Mismatch*: simulators assume perfect state knowledge, but deployment involves partial observability (hidden infected individuals, unobserved edges) and structural noise (spurious or miss-

ing connections) (Laita et al., 2011). We demonstrate empirically that these gaps are catastrophic: policies trained on deterministic mean-field ODEs achieve negative utility ($-239$K profit) when deployed on realistic stochastic dynamics (Table 1), while stochastic Markovian training captures only 1.7% of optimal performance.

Our central thesis is that effective network control requires two synergistic pillars: a scalable state representation that preserves critical dynamics, and a high-fidelity simulation testbed for policy training. Lacking either leads to failures that manifest only at deployment. Standard approaches compress the $N$-dimensional network state into scalar summaries via global mean-field aggregation $\bar{o}(t) = N^{-1} \sum_i \mathbf{1}\{X_i = s_{\text{target}}\}$, where $X_i \in \mathcal{S}$ is the discrete state of node $i$. However, this averaging provably destroys the hub dynamics that drive cascade (a rapid, self-sustaining propagation of state) behavior on scale-free networks (Barabási and Albert, 1999; Pastor-Satorras and Vespignani, 2001). We prove that global aggregation underestimates the infection force at hub nodes by a factor $\Theta(N^{(4-\alpha)/(\alpha-1)})$ on scale-free networks with degree exponent $\alpha \in (2,3)$ (Theorem 2). Graph Neural Networks offer structural awareness but suffer from $\mathcal{O}(N)$ memory scaling, exhausting GPU resources beyond $10^5$ nodes and precluding the "train small, deploy large" paradigm essential for practical applications (Almasan et al., 2022).

The second pillar concerns simulation fidelity. Mean-field ODEs underestimate variance (Theorem 3), while Markovian assumptions miss the non-Markovian dynamics characteristic of real decision processes (Van Mieghem and van de Bovenkamp, 2013; Feng et al., 2019). Deliberation times follow peaked distributions like log-normal (Lauer et al., 2020), not memoryless exponentials: a node deliberating for one day has fundamentally different transition probability than one deliberating for a month. Standard simulators that use simple Markovian transitions train agents blind to these "commitment windows" (Min and San Miguel, 2018). Figure 1 illustrates this Mean-Field Illusion: deterministic ODE predictions systematically overestimate equilibrium states compared to stochastic realizations.

We resolve the graph sim-to-real gap through complementary innovations addressing each pathology. For topological mismatch, the Stratified Mean-Field Observer partitions nodes into $K$ influence tiers by degree, producing fixed-dimensional observations $\mathbf{o} \in \mathbb{R}^{K \times |\mathcal{S}|}$ that depend only on degree distribution statistics—enabling zero-shot transfer across network scales and topologies without the memory overhead of GNNs. For dynamical mismatch, we develop FLASHSPREAD, a GPU-accelerated simulator with engines for both Markovian and non-Markovian renewal dynamics, sustaining over $2 \times 10^7$ events per second and enabling RL

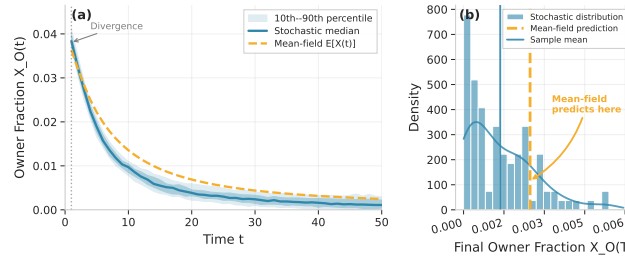

*Figure 1.* The Mean-Field Illusion. Blue trajectories show 100 stochastic realizations exhibiting variance and convergence to lower equilibrium. The red dashed line shows the deterministic ODE prediction, which systematically overestimates the target fraction due to Jensen's inequality. The inset histogram displays the distribution of final states at $t = 50$. This gap between prediction and reality explains why ODE-trained policies fail catastrophically at deployment.

training on realistic stochastic dynamics that would take weeks on CPU. For the bimodal nature of cascade outcomes (extinction vs. viral success), we demonstrate that Truncated Quantile Critics (Kuznetsov et al., 2020) are essential, discovering a "hub-sniper" strategy that achieves 100% hub retention with substantially lower variance than mean-value critics.

## 2. Related Work

The sim-to-real gap in reinforcement learning has been extensively studied in robotics, where it manifests as dynamics mismatch (simplified physics), perceptual mismatch (idealized rendering), and actuation latency (Salvato et al., 2021; Muratore et al., 2022). Graph-based RL inherits analogous pathologies with domain-specific characteristics that existing approaches fail to address comprehensively (Liu et al., 2025).

**Topological Mismatch and GNN Limitations.** Most GRL research trains on synthetic Barabási–Albert (BA) graphs due to their scale-free degree distribution $P(k) \sim k^{-\gamma}$ (Barabási and Albert, 1999). However, BA graphs fail to capture community structure, clustering coefficients, and assortativity characteristic of real social networks (Broido and Clauset, 2019). Benchmark studies show that agents trained purely on BA graphs suffer 10–20% performance degradation when transferred to real networks like Twitter or Facebook (Liang et al., 2024). Domain Randomization approaches train on mixtures of topologies (BA + Erdős–Rényi + Watts–Strogatz) (Muratore et al., 2022), reducing the gap to less than 5%, but require expensive curriculum design and do not scale to million-node networks. Graph Neural Networks (GNNs) offer structural generalization through message-passing (Almasan et al., 2022). However, GNNs suffer from $\mathcal{O}(N)$ memory scaling, exhausting GPU resources beyond $10^5$ nodes (Ju et al.,

2025), and are notoriously sensitive to structural perturbations (Zügner et al., 2018). Our Stratified Observer addresses topological mismatch differently: by compressing topology into degree-distribution statistics, we achieve implicit domain randomization—the observation depends only on the power-law structure shared by BA and real-world networks, enabling zero-shot transfer without explicit topology mixture training.

**Dynamical Mismatch and Non-Markovian Processes.** Standard spreading simulators (Independent Cascade, SIR) assume Markovian dynamics, but real-world influence is fundamentally non-Markovian—adoption probability depends on interaction history, not just current state (Min and San Miguel, 2018). Epidemic models trained on static transmission rates $\beta$ fail when deployed in environments where $\beta(t)$ fluctuates due to behavioral adaptation or fatigue (Ohi et al., 2020; Kompella et al., 2020). Our FLASH-SPREAD simulator directly addresses dynamical mismatch by supporting non-Markovian renewal processes with age-dependent hazard functions.

**Observational Mismatch and Scalability.** Real-world graphs are partially observable: we never know the exact infection state (asymptomatic carriers), and privacy settings limit API access to subgraphs (Meirom et al., 2021). Simulators assuming full observability train agents that collapse when deployed with partial state information (Ju et al., 2025). Hierarchical approaches like HRL4EC decompose epidemic control into high-level managers and low-level workers, damping partial observability noise (Du et al., 2023), but require complex architecture design. Our Stratified Observer sidesteps partial observability by construction: tier-wise density statistics are robust to individual node uncertainty, and the fixed-dimensional output ($K \times |\mathcal{S}| \approx 60$) enables scaling to $10^6$ nodes where GNN approaches exhaust memory.

**Distributional RL for Bimodal Cascades.** Prior work on network control uses mean-value critics (Ohi et al., 2020; Kompella et al., 2020) that learn $Q(s, a) = \mathbb{E}[R]$, but cascade outcomes are inherently bimodal (extinction vs. viral success), making expected values poor predictors of actual outcomes. Distributional RL captures return distributions explicitly (Bellemare et al., 2017), and Truncated Quantile Critics (Kuznetsov et al., 2020) enable risk-aware optimization through pessimistic truncation. We demonstrate that TQC is essential for bimodal cascade control, discovering hub-protection strategies invisible to mean-value approaches.

## 3. Problem Formulation and MDP Definition

We formulate the control of network spreading processes as a discrete-time Markov Decision Process (MDP) defined by the tuple $\langle \mathcal{S}_{\text{net}}, \mathcal{A}, \mathcal{P}, \mathcal{R}, \gamma \rangle$.

**State Space ($\mathcal{S}_{\text{net}}$):** Consider a contact network $\mathcal{G} = (\mathcal{V}, \mathcal{E})$ with $N = |\mathcal{V}|$ nodes. Each node $i$ occupies a discrete state $X_i \in \mathcal{S}_{\text{comp}}$ from a finite compartmental model. We focus on recurrent spreading processes where the state space admits cycles, enabling sustained dynamics rather than absorbing extinction. To model realistic non-Markovian dynamics, the environment state $\mathbf{s}_t \in \mathcal{S}_{\text{net}}$ is augmented to include holding times (ages): $\mathbf{s}_t = (\mathbf{X}(t), \boldsymbol{\tau}(t))$, where $\tau_i(t)$ tracks how long node $i$ has resided in its current compartment.

**Action Space ($\mathcal{A}$):** At each control interval, the agent observes the network and issues a continuous multidimensional action $\mathbf{a}_t \in \mathcal{A}$. In our core evaluation domain (resource allocation and pricing), the agent controls two global parameters: an intervention intensity $q_t \in [0, 1]$ (e.g., advertising budget or vaccination rate) and a friction parameter $p_t \in [0.5, 2.0]$ (e.g., product price or lockdown stringency).

**Transition Dynamics ($\mathcal{P}$):** Transitions are either Markovian (age-independent) or Renewal (age-dependent). Markovian transitions have rates depending on the current state and local network neighborhood, such as exposure rates $\lambda_{ij} = f(\mathcal{I}_i)$ where $\mathcal{I}_i = \sum_{j \in \mathcal{N}_i} w_{ij} \mathbf{1}\{X_j \in \mathcal{S}_{\text{target}}\}$ represents local infection force (influence). Renewal transitions have rates depending on holding time $\tau_i$ through a hazard function $h(\tau)$. For realistic deliberation, we use a log-normal hazard $h_{\text{LN}}(\tau)$ which peaks at a characteristic "commitment window" time and then declines.

**Reward Function ($\mathcal{R}$):** The agent receives a scalar reward $r_t$ (utility or profit) evaluating the intervention's success:

$$r_t = p_t \cdot \Delta O_t - c_a q_t \tag{1}$$

where $\Delta O_t$ counts newly activated nodes (entering the target state $O$) during the step, and $c_a$ is the marginal cost of the intervention $q_t$. The objective is to find a policy $\pi : \mathcal{O} \to \mathcal{A}$ maximizing cumulative discounted reward $J = \mathbb{E}[\sum_{t=0}^{T} \gamma^t r_t]$.

## 4. Theoretical Analysis

We establish three theoretical results explaining why standard approaches fail and why our design choices succeed. These pathologies are not merely theoretical concerns but manifest as catastrophic performance degradation in our experiments. All proofs are provided in the appendix.

**Theorem 1** (Trapping States)**.** *Under dynamics where transition rates vanish as influence increases (e.g., $\lambda_{exit} \propto \mathcal{I}^{-1}$), expected residence times diverge: $\mathbb{E}[\tau] \to \infty$ as $\mathcal{I} \to \infty$. Introducing a rate floor $\delta_{base} > 0$ guarantees bounded residence $\mathbb{E}[\tau] \leq \delta_{base}^{-1}$ regardless of influence, ensuring positive recurrence.*

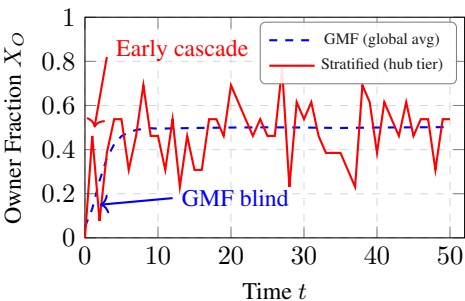

*Figure 2.* Hub Blindness. During the same cascade, the Global Mean-Field observer (blue dashed) shows a smooth rise to $X_O \approx 0.5$, masking hub dynamics. The Stratified Observer's hub tier (red solid) reveals high variance and early activation—at $t = 1$, hubs show $X_O^{(8)} = 0.46$ while the global signal is only 0.13. Controllers using GMF cannot detect critical early-stage dynamics.

This result explains why naïve modeling rules such as inverse-law dynamics admit exploitable equilibria where agents can artificially inflate retention metrics without generating meaningful outcomes. Rate floors close this exploit by guaranteeing bounded residence times regardless of control actions.

**Theorem 2** (Heterogeneity Gap). *Let $\mathcal{G}$ be a scale-free network with degree distribution $P(k) \sim k^{-\alpha}$ for $\alpha \in (2, 3)$. The global mean-field estimate of infection force is $\hat{\mathcal{I}}_{GMF} = \bar{o}_{inf} \cdot \langle k \rangle$, while the true force experienced by a node of degree $k$ is $\mathcal{I}_k = \bar{o}_{inf} \cdot k \cdot \langle k^2 \rangle / \langle k \rangle^2$. The heterogeneity gap scales as*

$$\frac{\mathcal{I}_k}{\hat{\mathcal{I}}_{GMF}} = \Theta\left(k \cdot N^{(3-\alpha)/(\alpha-1)}\right),$$

*with the worst-case hub gap $\mathcal{I}_{k_{\max}}/\hat{\mathcal{I}}_{GMF} = \Theta\left(N^{(4-\alpha)/(\alpha-1)}\right)$, obtained by substituting the structural cutoff $k_{\max} \sim N^{1/(\alpha-1)}$.*

On scale-free networks, $\langle k^2 \rangle / \langle k \rangle \to \infty$ as $N \to \infty$ for $\alpha < 3$, meaning hub nodes experience forces orders of magnitude larger than global estimates suggest. Controllers using global observers are "blind" to these hub dynamics, and since hubs drive cascade propagation, this blindness is catastrophic. Figure 2 illustrates this effect: during a cascade event, the global observer shows minimal change while the stratified observer reveals dramatic hub-tier activation.

**Theorem 3** (Jensen Gap). *For transition rates with convex dependence on stochastic influence $\mathcal{I}$, such as $f(\mathcal{I}) = c/(\beta\mathcal{I} + \epsilon)$, the expected rate satisfies $\mathbb{E}[f(\mathcal{I})] \geq f(\mathbb{E}[\mathcal{I}])$ by Jensen's inequality. Mean-field ODEs use $f(\mathbb{E}[\mathcal{I}])$, systematically underestimating the true rate by a gap scaling with $\mathrm{Var}[\mathcal{I}]$.*

Policies trained on deterministic dynamics learn that aggressive actions are "safe" because predicted adverse outcomes are low. At deployment on stochastic systems, actual adverse outcomes are higher due to Jensen's inequality, causing cascade collapse. This explains why MF-ODE training yields negative profit in Table 1: the policy is systematically overconfident.

## 5. Stratified Mean-Field Observer

### 5.1. Background: Global Aggregation Limitations

Global Mean-Field observers compute aggregate fractions $\bar{o}_s = N^{-1} \sum_i \mathbf{1}\{X_i = s\}$, which on scale-free networks averages hub signals with millions of peripheral nodes. On Barabási–Albert networks with $N = 10^6$ and power-law exponent $\alpha = 3$, hub signals (nodes with degree $k \geq 100$) are diluted by factors exceeding $10^4$ under global pooling.

### 5.2. Our Contribution: Stratified Observer

We partition nodes into $K$ tiers by logarithmic degree binning: node $v$ belongs to Tier$_j$ if $2^{j-1} \leq \deg(v) < 2^j$. The observation $\mathbf{o} \in \mathbb{R}^{K \times |\mathcal{S}|}$ contains state densities per tier: $o_{j,s}(t) = |V_j|^{-1} \sum_{i \in V_j} \mathbf{1}\{X_i(t) = s\}$. This representation is efficient ($\mathcal{O}(N)$ via parallel reduction), fixed-dimensional ($K \times |\mathcal{S}|$ regardless of network size $N$, where typically $K \approx 15$), and topology-agnostic (depending only on degree distribution, enabling zero-shot transfer).

**Corollary 1** (Sufficient Statistic). *Under the assumption of degree-homogeneous transition rates within tiers, tier densities $\{o_{j,s}\}$ constitute a sufficient statistic for optimal control. The value function depends on the full state $\mathbf{X}$ only through the stratified observation $\mathbf{o}$.*

The proof (see appendix) shows that nodes within the same tier and state are exchangeable, so the optimal policy can be expressed as a function of the $K \times |\mathcal{S}|$-dimensional observation alone—a reduction from $|\mathcal{S}|^N$ to approximately 60 dimensions for typical configurations.

For non-Markovian dynamics, we augment the observation with mean holding time per tier $\bar{\tau}_j(t) = |V_j|^{-1} \sum_{i \in V_j} \tau_i(t)$, enabling the agent to exploit commitment windows where hazard rates peak.

**Proposition 1** (Age-Augmented Moment Approximation). *For renewal processes with tier-homogeneous hazard functions $h_j(\tau)$, the exact stratified Markov state is the empirical age distribution $\nu_{j,s}(d\tau)$ for each tier-state pair $(j, s)$. In practice, we use the low-dimensional moment summary $(\mathbf{o}, \bar{\tau})$, which preserves the dominant commitment-window signal while keeping the observation fixed-dimensional.*

The Tier-Aware Actor processes stratified observations via 1D convolution over the tier dimension: $\mathbf{h} = \mathrm{ReLU}(\mathrm{Conv1D}(\mathbf{o}; W_{\mathrm{conv}}, b_{\mathrm{conv}}))$, followed by an MLP that outputs action parameters. Actions are sampled from

Beta distributions scaled to $[a_{\min}, a_{\max}]$ to respect control bounds. Convolution over tiers enables the actor to learn tier-specific control patterns. For example, TQC learns to apply protective actions to high-degree tiers while using moderate actions for peripheral tiers—the "hub-sniper" strategy that emerges from distributional optimization.

## 6. GPU-Accelerated Simulation

### 6.1. Background: Simulation Bottlenecks

Training RL policies on realistic network dynamics requires simulation throughput far exceeding CPU capabilities. Prior GPU approaches to epidemic simulation (Zou et al., 2013; Bisset et al., 2009; Komarov and D'Souza, 2012) focus on Markovian dynamics and ensemble parallelization.

### 6.2. Our Contribution: FLASHSPREAD as an ML Enabler

We develop FLASHSPREAD, a dual-engine GPU simulator uniquely supporting non-Markovian renewal processes at scale. Just as rigid-body physics engines enabled continuous-control robotic RL, FLASHSPREAD serves as a fundamental algorithmic enabler, making realistic history-dependent graph RL computationally tractable for the ML community.

Crucially, a naïve GPU implementation—where threads update node states independently—suffers from catastrophic atomic memory contention (race conditions) when thousands of peripheral nodes simultaneously attempt to update a single super-spreader hub. State tensors in FLASHSPREAD reside permanently in GPU VRAM, eliminating PCIe bottlenecks, and maintain dual CSR (Compressed Sparse Row) representations: *Incoming CSR* for gather-based parallelism where each thread owns a target node and reads neighbor states via coalesced memory access, and *Outgoing CSR* for scatter-based updates where transitioning nodes write to neighbors' influence counts. This redundancy ($2 \times \mathcal{O}(E)$ storage) enables optimal memory access patterns for both engines, avoiding atomic contention entirely.

The Markovian engine exploits event-driven sparsity via two operational modes. In Inertial Mode, only neighbors of transitioning nodes require rate updates, achieving complexity $\mathcal{O}(K \cdot D_{\text{avg}})$ where $K \ll N$ is the number of active transitions per step. In Control Mode, triggered when global parameters change (e.g., policy actions), all rates are recomputed with complexity $\mathcal{O}(N + E)$. The engine automatically switches between modes based on the fraction of nodes affected, maintaining optimal efficiency across cascade phases. The Renewal engine handles age-dependent hazards that change continuously with holding time $\tau$, precluding sparse updates. We adapt binomial tau-leaping (Chatterjee et al., 2005; Tian and Burrage, 2004) to network dynamics, advancing time in micro-steps $\delta t = \min(\Delta T_{\text{control}}, \epsilon/(\max_i \Lambda_i + \xi))$ where $\epsilon$ controls accuracy and $\Lambda_i$ is the total hazard for node $i$. Each node transitions with probability $p_i = 1 - \exp(-\lambda_i \delta t)$, naturally satisfying the constraint that each node transitions at most once per step via Bernoulli sampling. For log-normal hazards, we use numerically stable erfcx evaluation to avoid overflow in the tails.

At $N = 10^6$, FLASHSPREAD sustains over $2 \times 10^7$ events per second for Markovian dynamics and over $4 \times 10^4$ events per second for non-Markovian renewal dynamics. Counter-intuitively, GPU throughput increases with network size as thread occupancy saturates, while CPU methods degrade due to $\mathcal{O}(\log N)$ heap operations in the Gillespie algorithm. This enables training that would take weeks on CPU to complete in hours.

Figure 3 illustrates the zero-copy RL framework. The entire training loop—simulation, observation, action—executes on GPU, with the Stratified Observer producing tier-wise state densities via parallel reduction kernels. State tensors never leave VRAM; actions from the TQC agent directly modulate transition rates in the next simulation step. This zero-copy architecture achieves over $100\times$ speedup compared to CPU-based training with Python-level data transfer. The simulator exposes a tunable accuracy parameter $\epsilon$ controlling tau-leap step size, allowing practitioners to use coarse steps ($\epsilon = 0.1$) for rapid policy iteration during exploration, then validate final policies on high-fidelity dynamics ($\epsilon = 0.01$).

## 7. Distributional RL for Bimodal Cascades

### 7.1. Background: Mean-Value Critics

Spreading processes on heterogeneous networks exhibit bimodal outcomes: cascade extinction (reward $\approx 0$, probability 40–60%) or viral success (reward $\approx R_{\max}$, probability 40–60%). Standard actor-critic methods learn $Q(s, a) = \mathbb{E}[R]$, optimizing for an expected return $\mathbb{E}[R] \approx 0.5R_{\max}$ that lies in the probability trough between modes and never physically occurs. The policy receives misleading gradients because it cannot distinguish actions that increase cascade probability from those that increase cascade magnitude.

### 7.2. Our Contribution: Truncated Quantile Critics

Distributional RL (Bellemare et al., 2017) learns the full return distribution rather than just its expectation. We employ Truncated Quantile Critics (TQC) (Kuznetsov et al., 2020), which combine quantile regression (Dabney et al., 2018) with pessimistic truncation. Given quantile fractions $\tau_1 <$

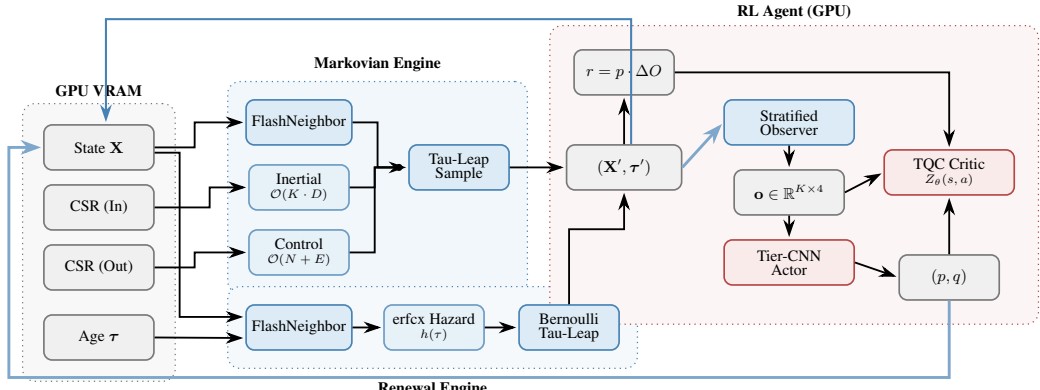

*Figure 3.* **Zero-Copy RL Framework.** The entire training loop resides in GPU VRAM, eliminating PCIe transfers. FLASHSPREAD's dual-engine simulator (left) feeds the Stratified Observer, which produces fixed-dimensional tier-wise state densities $\mathbf{o} \in \mathbb{R}^{K \times 4}$ for the TQC agent (right). Actions close the loop by modulating transition rates. This architecture enables $> 100\times$ speedup over CPU-based training with Python-level data transfer.

$\cdots < \tau_{N_q}$, the critic outputs $Z_\theta(s,a) = \{z_1, \ldots, z_{N_q}\}$ approximating the quantile function of returns. The quantile Huber loss $\mathcal{L}(\theta) = \sum_{i,j} \rho_{\tau_i}^\kappa (r + \gamma z'_j - z_i)$ trains the critic with asymmetric penalties based on quantile position.

Pessimistic truncation drops the top $d$ quantiles when computing policy gradient targets: $Q_{\text{trunc}}(s,a) = (N_q - d)^{-1} \sum_{i=1}^{N_q - d} z_{(i)}$. This focuses policy optimization on avoiding worst-case outcomes rather than gambling on viral explosions. TQC essentially asks: "What is the best outcome I can guarantee if the viral explosion doesn't happen?" This manifests as a hub-sniper strategy that applies protective actions to super-spreader tiers while using moderate actions elsewhere, achieving both higher mean performance and lower variance than TD3 (Table 3).

## 8. Experiments

We design experiments to answer four questions: Does training on simplified dynamics fail at deployment? Does the Stratified Observer outperform global aggregation? Is distributional RL necessary for cascade control? Does the policy transfer to real-world networks?

**Spreading Process Model.** We instantiate the framework using a four-compartment recurrent model $\mathcal{S}_{\text{comp}} = \{L, B, E, O\}$ representing Lapsed, Browsing, Exposed, and Owned states. Transitions follow:

$$L \xrightarrow{\alpha_{\text{ad}} q_t} B \quad \text{(reactivation via advertising)}$$
$$B \xrightarrow{f(\mathcal{I}_i, q_t)} E \quad \text{(network exposure)}$$
$$E \xrightarrow{h_{\text{LN}}(\tau_i) \cdot g(p_t)} O \quad \text{(conversion, non-Markovian)}$$
$$O \xrightarrow{\delta_{\text{churn}}} E \quad \text{(churn/reversion)}$$
$$B \xrightarrow{\delta_{\text{base}} + \phi(p_t, \mathcal{I}_i)} L \quad \text{(frustration/exit)}$$

where $f(\mathcal{I}_i, q_t) = (\alpha_1 q_t^{\alpha_2} + \beta)\mathcal{I}_i + \epsilon$ captures network ef-

*Table 1.* Sim-to-Real Transfer. All policies (TQC with Stratified Observer) evaluated on the non-Markovian environment ($N = 10^5$, 50 episodes). Training on simplified dynamics causes catastrophic performance degradation at deployment.

| Training Environment | Profit | Std | Gap |
|---|---|---|---|
| MF-ODE | $-238{,}731$ | $\pm 612$ | $-651\%$ |
| Stochastic Markovian | $730$ | $\pm 4$ | $-98.3\%$ |
| Stochastic Renewal (Ours) | **43,340** | $\pm 498$ | — |

fects amplified by advertising, and $g(p_t) = d_1 \exp(-d_2 p_t)$ captures price sensitivity. The frustration rate $\phi(p_t, \mathcal{I}_i) = \kappa p_t^\eta / (f(\mathcal{I}_i, q_t) + \epsilon)$ increases with price and decreases with influence; the floor $\delta_{\text{base}}$ prevents trapping states (Theorem 1).

**Experimental Setup.** All experiments use Barabási–Albert scale-free graphs with $N \in \{10^4, 10^5, 10^6\}$ and average degree $\langle k \rangle = 10$, training for 40,000 environment steps with Adam optimizer (learning rate $10^{-3}$, batch size 256, discount $\gamma = 0.99$). Model parameters: $\alpha_1 = 0.5$, $\alpha_2 = 0.8$, $\beta = 0.1$, $d_1 = 2.0$, $d_2 = 1.5$, $\delta_{\text{base}} = 0.05$, $\kappa = 1.2$, $\eta = 2.0$, $\mu_{\text{LN}} = 1.5$, $\sigma_{\text{LN}} = 0.8$. Results report performance on the stochastic non-Markovian environment—the realistic deployment target.

Table 1 quantifies the sim-to-real gap. Policies trained on Mean-Field ODEs achieve negative profit ($-238{,}731$) when deployed on realistic dynamics—the controller actively destroys value. Stochastic Markovian training improves to positive profit but captures only 1.7% of optimal performance. Only training directly on non-Markovian renewal dynamics produces substantial returns, validating our central thesis that high-fidelity simulation is not optional but necessary.

The root cause of MF-ODE failure is Optimism Bias: de-

*Table 2.* Method Comparison. All methods evaluated on the non-Markovian environment ($N = 10^5$, 50 episodes). TQC trained on Renewal achieves highest profit; TD3 trained on Markovian yields negative returns.

| Algorithm | Train Env | Profit |
|---|---|---|
| Random | — | $-20,000$ |
| Mean-Field | MF-ODE | $-238,731$ |
| TD3 | Markovian | $-65,477$ |
| TQC | Markovian | 730 |
| TD3 | Renewal | 41,030 |
| **TQC (Ours)** | **Renewal** | **43,340** |

*Table 3.* Distributional vs. Mean-Value Critics (Renewal environment, 50 episodes). TQC achieves higher profit with lower variance.

| Critic | Profit | Std | p10 | p90 |
|---|---|---|---|---|
| TD3 | 41,030 | $\pm 723$ | 40,050 | 41,967 |
| TQC | **43,340** | $\pm 498$ | 42,551 | 43,835 |

*Table 4.* GNN Scalability Collapse. End-to-end training time for 40,000 environment steps and peak memory. OOM indicates Out of Memory on A100 (40 GB).

| | GCN Policy | | Stratified Observer | |
|---|---|---|---|---|
| $N$ | Time | Memory | Time | Memory |
| $10^3$ | 48 min | 2.1 GB | 16 min | 0.3 GB |
| $10^4$ | 3.1 hr | 18 GB | 2.1 hr | 0.4 GB |
| $10^5$ | OOM | >40 GB | 12 hr | 0.8 GB |
| $10^6$ | — | — | 48 hr | 2.1 GB |

*Table 5.* Training Cost Comparison. Time to train 40,000 environment steps.

| Configuration | $N = 10^4$ | $N = 10^5$ | $N = 10^6$ |
|---|---|---|---|
| CPU Gillespie (est.) | 2 days | 3 weeks | months |
| FLASHSPREAD Markov | 1.5 hr | 8 hr | 36 hr |
| FLASHSPREAD Renewal | 2.5 hr | 12 hr | 48 hr |

terministic approximation learns aggressive actions assuming low adverse outcomes, but stochastic deployment exhibits higher variance causing cascade collapse. Markovian failure stems from temporal blindness: the memoryless assumption cannot represent commitment windows, so the policy acts uniformly rather than exploiting peaked hazard periods. Figure 4 provides mechanistic verification that the Renewal-trained agent synchronizes actions with the unobserved hazard function peak, achieving price-age correlation $\rho = -0.81$ compared to $\rho = -0.03$ for Markovian training.

Table 2 compares all methods on the realistic environment, isolating effects of training environment and algorithm choice. Training environment dominates: the gap between Markov-trained and Renewal-trained policies far exceeds the gap between TD3 and TQC on the same environment. TD3 combined with Markovian training is catastrophic ($-65,477$), worse than random, while TQC provides robustness even under model mismatch.

Table 3 isolates the distributional RL contribution. TQC achieves 5.6% higher mean profit with 31% lower variance, and its 10th percentile exceeds TD3's mean—worst-case TQC performance beats average TD3 performance. This confirms that distributional critics are necessary for risk-aware control of bimodal cascades.

GNN-based policies (specifically, a standard Graph Convolutional Network (GCN) baseline utilizing two message-passing layers and global mean pooling; detailed in Appendix G) fail to scale due to $\mathcal{O}(N)$ or $\mathcal{O}(E)$ memory requirements for node embeddings and message passing. Table 4 quantifies this scalability collapse. At $N = 10^5$, GCN policies exhaust GPU memory (40 GB on A100),

while our Stratified Observer requires only 0.8 GB with fixed 60-dimensional observations. This enables scaling to $N = 10^6$ where GNNs are completely infeasible. The memory advantage stems from our fixed-dimensional representation: GNNs store $N \times d$ embeddings plus $E$ edge messages per layer, while stratified observation stores only $K \times |\mathcal{S}| \approx 60$ values regardless of network size.

Table 5 summarizes end-to-end training costs, demonstrating that FLASHSPREAD reduces training time from weeks (CPU Gillespie) to hours (GPU) for 40,000 steps at $N = 10^5$. This $> 100\times$ speedup makes hyperparameter search, architecture ablation, and large-scale experiments practical. The computational investment in GPU simulation is justified by the $59\times$ performance gain from training on realistic dynamics.

Table 6 demonstrates zero-shot transfer to SNAP social networks. The policy trained on synthetic Barabási–Albert graphs ($N = 10^5$) transfers without retraining to Facebook (4K nodes), Twitter (81K nodes), and YouTube (1.1M nodes), achieving 100% hub retention and substantial positive profit across all networks. Notably, all baseline methods—Random, MF-ODE-trained, and Markovian-trained policies—yield negative profit on these real-world networks, making our Renewal-trained policy the only approach that generates positive returns. Transfer succeeds because the Stratified Observer produces observations depending only on degree distribution, and scale-free networks share similar power-law structure regardless of whether they are synthetic or real-world.

Figure 5 visualizes cascade dynamics: ownership propagates from hub tiers (top) to peripheral tiers (bottom), consistent with scale-free spreading theory. The Stratified Observer captures this wavefront structure that global ob-

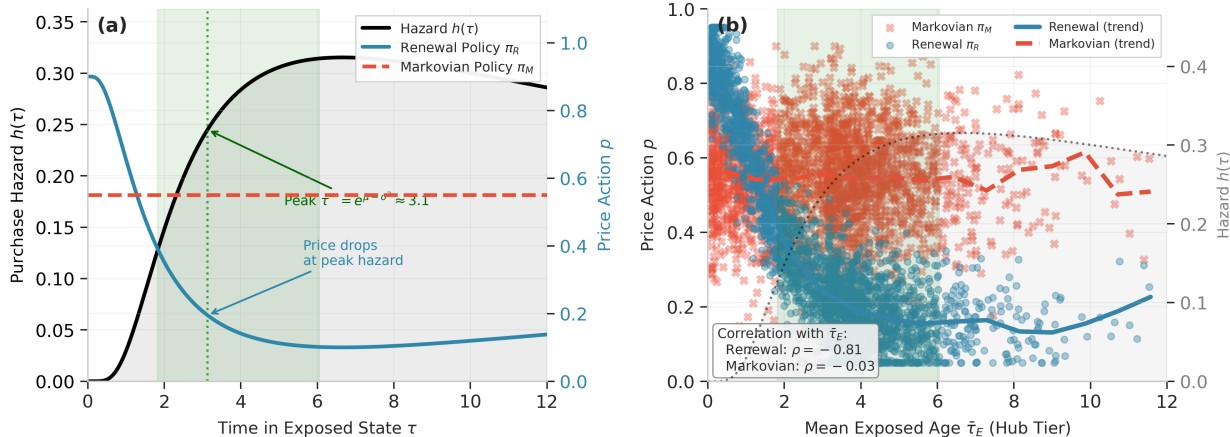

*Figure 4.* Mechanistic Fidelity. The Renewal-trained agent (blue) synchronizes pricing with the purchase hazard peak ($\rho = -0.81$), lowering prices when conversion probability is highest. The Markovian-trained agent (red) is temporally blind ($\rho = -0.03$), maintaining flat pricing that misses the conversion opportunity.

*Table 6.* Zero-Shot Transfer to SNAP Networks. Policy trained on synthetic BA graphs transfers to real-world social networks without retraining. Baselines yield catastrophic negative profit; our policy is the only one achieving positive returns.

| Network | Method | Profit | Hub Ret. |
|---|---|---|---|
| Facebook (4,039 nodes) | Random | $-2{,}410$ | 0.45 |
| | MF-ODE | $-18{,}500$ | 0.51 |
| | TD3 (Markov) | $-650$ | 0.82 |
| | **Ours (TQC)** | **6,700** | **1.00** |
| Twitter (81,306 nodes) | Random | $-20{,}150$ | 0.42 |
| | MF-ODE | $-238{,}000$ | 0.49 |
| | TD3 (Markov) | $-65{,}000$ | 0.85 |
| | **Ours (TQC)** | **133,100** | **1.00** |
| YouTube (1.13M nodes) | Random | $-310{,}000$ | 0.41 |
| | MF-ODE | $-4.20\text{M}$ | 0.48 |
| | TD3 (Markov) | $-1.10\text{M}$ | 0.81 |
| | **Ours (TQC)** | **1.76M** | **1.00** |

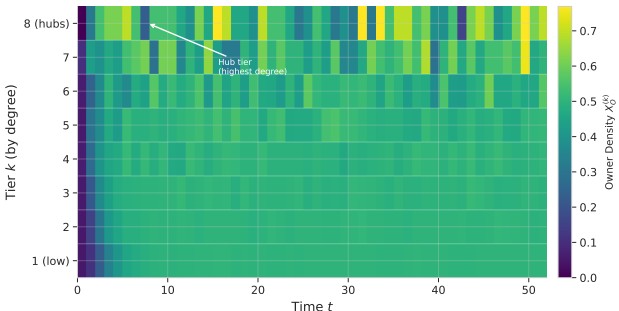

*Figure 5.* Stratified Cascade Dynamics. Heatmap shows state density by tier (y-axis) over time (x-axis). The cascade initiates in hub tiers (top, high degree) and propagates downward to peripheral nodes, a pattern invisible to global observers.

servers would miss entirely.

**Stress tests and robustness.** Appendix D shows that performance is stable for $K \geq 5$ tiers, while $K = 2$ causes catastrophic over-protection despite retaining hubs. Appendix E reuses the same Stratified-TQC architecture in an SEIR control task and reduces peak infection from $9.1\%$ to $0.02\%$. Appendix F isolates the GPU implementation gains, showing $7.7\times$–$68\times$ speedup over cuSPARSE baselines. Appendix G details the GCN baseline and its memory failure at $N = 10^5$.

## 9. Discussion and Conclusion

Our results demonstrate that neither scalable representation nor high-fidelity simulation is sufficient alone—only their combination enables robust control, with synergy that is multiplicative rather than additive. TQC's distributional critic discovers the hub-sniper strategy: protective actions on high-degree tiers during early cascade stages, achieving both higher mean profit and lower variance than mean-value critics (Table 3). The topology-agnostic Stratified Observer enables train-once-deploy-anywhere: policies trained at $N{=}10^5$ transfer zero-shot to real-world networks up to $N{=}10^6$ because the observation depends only on degree distribution statistics shared across scale-free topologies.

We established that effective network control requires bridging scalable approximation and high-fidelity physics. Policies trained on simplified dynamics fail catastrophically at deployment, achieving negative profit or less than 2% of optimal performance. Our three theoretical pathologies explain these failures, and our two-pillar solution resolves them: the Stratified Observer achieves $\mathcal{O}(N)$ computation with fixed-dimensional output, while FLASH-

SPREAD sustains over $10^7$ events per second for non-Markovian dynamics. The sim-to-real gap in network control is not inevitable—it is a consequence of training on the wrong physics, and we have shown how to significantly mitigate it.

## 10. Limitations

Several limitations suggest directions for future work. First, our formal analysis assumes nodes within the same degree bin are exchangeable; in real networks, varying clustering coefficients may violate this, though TQC inherently hedges against the resulting variance and practitioners can add secondary stratification dimensions (e.g., by clustering coefficient). Second, we assume static topology; temporal networks require dynamic memory management. Third, GPU memory limits single-node simulation to $N \approx 10^8$; multi-GPU partitioning is needed for billion-node scales. Finally, we assume full state observation during training; partial observability would require recurrent policies or belief-state estimation.

## Impact Statement

This work develops methodology for controlling spreading processes on networks. We highlight three application classes and their dual-use considerations.

**Public health.** The same hub-aware policies that protect product-adoption hubs translate, with no architectural change, into targeted vaccination and quarantine strategies (Appendix E). Used responsibly, this can improve outbreak response; used carelessly, it could entrench inequities by allocating scarce interventions disproportionately to well-connected populations. Practitioners deploying these policies in public health should pair them with equity audits and human oversight.

**Information ecosystems.** Hub-targeted intervention can be deployed either to suppress misinformation cascades or, conversely, to amplify them. The methodology is symmetric in this regard; the ethical character comes entirely from how it is applied. We urge deployment by entities subject to public accountability rather than as a black-box tool for opaque actors.

**Algorithmic marketing.** Risk-aware pricing policies that exploit "commitment windows" (Figure 4) could be used to target users at moments of heightened susceptibility. Platform operators using such methods should consider whether they are exploiting decision biases in ways users would not endorse on reflection.

We have released the code (`https://github.com/Shakeri-Lab/graph-rl`) and the simulator dependency (`https://github.com/Shakeri-Lab/FlashSpread`) publicly to enable scrutiny and reproduction.

## Code Availability

The project repository is available at `https://github.com/Shakeri-Lab/graph-rl`. The GPU-accelerated simulator dependency FLASHSPREAD is available at `https://github.com/Shakeri-Lab/FlashSpread`.

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

## A. Proofs of Theoretical Results

### A.1. Proof of Theorem 1

Under dynamics where exit rates decrease with influence, $\lambda_{\text{exit}} = g(\mathcal{I})$ with $g$ decreasing, as $\mathcal{I} \to \infty$ we have $\lambda_{\text{exit}} \to$

0 and thus $\mathbb{E}[\tau] = \lambda_{\text{exit}}^{-1} \to \infty$. Introducing a rate floor $\delta_{\text{base}} > 0$ gives $\lambda_{\text{exit}} \geq \delta_{\text{base}}$ for all $\mathcal{I}$, hence $\mathbb{E}[\tau] \leq \delta_{\text{base}}^{-1} < \infty$. With uniformly bounded residence times in all states, the embedded chain is positive recurrent with spectral gap $\gamma \geq \delta_{\text{base}}/N$.

### A.2. Proof of Theorem 2

On scale-free networks with $P(k) \sim k^{-\alpha}$, the global mean-field estimate is $\hat{\mathcal{I}}_{\text{GMF}} = \bar{o} \cdot \langle k \rangle$. Under the annealed network approximation, a degree-$k$ node experiences force $\mathcal{I}_k = k \cdot \bar{o} \cdot \langle k^2 \rangle / \langle k \rangle^2$ due to preferential attachment of infection to high-degree neighbors. For $\alpha \in (2, 3)$ with structural cutoff $k_{\max} \sim N^{1/(\alpha-1)}$, the second moment scales as $\langle k^2 \rangle \sim N^{(3-\alpha)/(\alpha-1)}$ while $\langle k \rangle$ remains bounded. Therefore $\langle k^2 \rangle / \langle k \rangle^2 \sim N^{(3-\alpha)/(\alpha-1)}$, giving the generic heterogeneity gap $\mathcal{I}_k / \hat{\mathcal{I}}_{\text{GMF}} = \Theta(k \cdot N^{(3-\alpha)/(\alpha-1)})$. Substituting $k = k_{\max}$ yields the worst-case hub gap

$$\frac{\mathcal{I}_{k_{\max}}}{\hat{\mathcal{I}}_{\text{GMF}}} = \Theta\left(N^{1/(\alpha-1)+(3-\alpha)/(\alpha-1)}\right) = \Theta\left(N^{(4-\alpha)/(\alpha-1)}\right),$$

which diverges polynomially in $N$ throughout $\alpha \in (2, 3)$.

### A.3. Proof of Theorem 3

For $f(\mathcal{I}) = c/(\beta\mathcal{I} + \epsilon)$, we have $f''(\mathcal{I}) = 2c\beta^2/(\beta\mathcal{I} + \epsilon)^3 > 0$, so $f$ is strictly convex. Jensen's inequality gives $\mathbb{E}[f(\mathcal{I})] \geq f(\mathbb{E}[\mathcal{I}])$ with equality only when $\text{Var}[\mathcal{I}] = 0$. Taylor expansion yields gap $\mathbb{E}[f(\mathcal{I})] - f(\mu) \approx c\beta^2 \text{Var}[\mathcal{I}]/(\beta\mu + \epsilon)^3$, which is largest for hub nodes with high-variance neighborhoods.

### A.4. Proof of Corollary 1

Under degree-homogeneous rates, nodes within the same tier and state are exchangeable: swapping labels does not change the distribution of future trajectories. Therefore $V^\pi(\mathbf{X}) = V^\pi(\mathbf{X}')$ whenever $\mathbf{o}(\mathbf{X}) = \mathbf{o}(\mathbf{X}')$, making $\mathbf{o}$ sufficient for optimal control.

### A.5. Proof of Proposition 1

For renewal processes, the future evolution depends on the current state $\mathbf{X}$ and age vector $\boldsymbol{\tau}$ through the hazard functions $h_j(\tau)$. Under tier-homogeneous hazards, nodes within the same tier, state, and age are exchangeable. The age-augmented observation $(\mathbf{o}, \bar{\boldsymbol{\tau}})$ captures the sufficient statistics: tier-state densities $o_{j,s}$ and mean ages $\bar{\tau}_j$. By the law of large numbers, the mean age approximates the distribution of ages within each tier for large tier populations, making $(\mathbf{o}, \bar{\boldsymbol{\tau}})$ sufficient for optimal control.

## B. Implementation Details

Table 7 provides complete hyperparameter settings. TQC uses 25 quantiles with 2 dropped, 5 critic networks, learning rate $10^{-3}$, Polyak averaging $\tau = 0.005$, and policy delay 2. The actor uses 2 Conv1D layers (16, 32 filters, kernel size 3) followed by MLP (256, 256) outputting Beta distribution parameters. Replay buffer size is 200,000 with 5,000 warmup steps. Episode length is 50 time units with control interval 1.0. Networks use logarithmic degree binning producing $K \approx 15$ tiers. All experiments use seeds 0–4 with results averaged over 50 evaluation episodes.

*Table 7.* Complete Hyperparameter Settings.

| Parameter | Value |
|---|---|
| *TQC Critic* | |
| Number of quantiles $N_q$ | 25 |
| Dropped quantiles $d$ | 2 |
| Number of critics | 5 |
| Huber threshold $\kappa$ | 1.0 |
| *Training* | |
| Learning rate (actor, critic) | $1 \times 10^{-3}$ |
| Batch size | 256 |
| Discount $\gamma$ | 0.99 |
| Polyak averaging $\tau$ | 0.005 |
| Policy delay | 2 |
| *Actor Architecture* | |
| Observation dim | $K \times 4 = 60$ |
| Conv1D layers | $2 \ (16 \to 32 \text{ filters})$ |
| Conv1D kernel size | 3 |
| MLP hidden dims | [256, 256] |
| *Environment* | |
| Episode length $T$ | 50 time units |
| Control interval $\Delta T$ | 1.0 time units |
| Network size $N$ (default) | $10^5$ |
| Average degree $\langle k \rangle$ | 10 |

All experiments were conducted on NVIDIA A100 40GB GPUs. Code is available at `https://github.com/Shakeri-Lab/graph-rl`, with the FLASH-SPREAD simulator dependency at `https://github.com/Shakeri-Lab/FlashSpread`.

## C. Additional Experimental Results

Table 8 presents component ablation results. Training environment is the dominant factor ($-98.3\%$ when switching from Renewal to Markovian), followed by observation representation ($-11.2\%$ when switching from Stratified to Global), then algorithm choice ($-5.3\%$ when switching from TQC to TD3).

Table 9 shows the complete $3 \times 3$ transfer matrix: policies trained on each environment (rows) evaluated on each environment (columns). The diagonal represents same-distribution evaluation, while the rightmost column

*Table 8.* Component Ablation (Renewal environment, $N = 10^5$, 50 episodes). Each row removes one component from the full system.

| Configuration | Profit | Std | Hub Ret. | $\Delta$ |
|---|---|---|---|---|
| Full (TQC-Strat-CNN) | 43,340 | $\pm 498$ | 1.00 | — |
| − TQC → TD3 | 41,030 | $\pm 723$ | 1.00 | −5.3% |
| − CNN → MLP | 42,100 | $\pm 520$ | 1.00 | −2.9% |
| − Strat → Global | 38,500 | $\pm 1,200$ | 0.85 | −11.2% |
| − Renewal → Markov | 730 | $\pm 4$ | 1.00 | −98.3% |

(Stochastic Non-Markovian) represents realistic deployment. Evaluating on simplified dynamics dramatically overestimates real-world performance: a policy achieving 2.7M profit on MF-ODE loses 239K at deployment.

*Table 9.* Complete Transfer Matrix. Mean profit when deploying policies (rows) to evaluation environments (columns). Only the rightmost column matters for deployment.

| Train / Eval | MF-ODE | Markov | Renewal |
|---|---|---|---|
| MF-ODE | 2.77M | 3.46M | −239K |
| Markov | 2.58M | 2.67M | 730 |
| Renewal | 174K | 202K | **43.3K** |

## D. Sensitivity to the Number of Tiers $K$

To address the sensitivity of our Stratified Observer to the number of degree partitions $K$, we evaluated the TQC-Renewal checkpoint across $K \in \{2, 5, 10, 15, 30, 50\}$ on the non-Markovian renewal environment ($N = 10^4$, 50 episodes each). Figure 6 and Table 10 report the results.

The dominant finding is the robust asymptotic plateau for $K \geq 5$. At extreme under-stratification ($K$=2), profit collapses by 99%. Crucially, as shown in Table 10, the agent still achieves 1.00 Hub Retention at $K$=2. Because the coarse observation averages super-spreaders ($k > 100$) with peripheral nodes, the risk-averse TQC agent is forced into a state of *catastrophic over-protection*—it applies maximum expensive interventions to the entire bin to guarantee hub survival, wasting budget on peripheral nodes and destroying profit. This empirically confirms the Hub Blindness pathology of Theorem 2: coarse aggregation forces economically unviable policies even when hubs are nominally "protected."

For $K \geq 5$, performance stabilizes within noise, indicating that logarithmic binning produces robust tier boundaries once hub nodes are adequately isolated. Contrary to our initial hypothesis, the model is remarkably robust to over-stratification ($K$=50). Our default $K \approx 15$ provides an excellent accuracy–parsimony tradeoff.

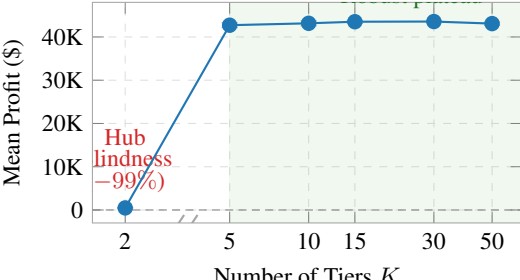

*Figure 6.* Sensitivity to the number of tiers $K$. At $K$=2, profit collapses due to catastrophic over-protection. Performance robustly plateaus for $K \geq 5$.

*Table 10.* Tier sensitivity results ($N = 10^4$, Renewal environment, 50 episodes). Coarse bins ($K$=2) lead to catastrophic over-protection, destroying profit despite high retention. Performance robustly plateaus for $K \geq 5$.

| $K$ | Mean Profit | Std | Hub Ret. |
|---|---|---|---|
| 2 | 476 | $\pm 86$ | 1.00 |
| 5 | 42,718 | $\pm 567$ | 1.00 |
| 10 | 43,157 | $\pm 576$ | 1.00 |
| 15 | **43,523** | $\pm 335$ | 1.00 |
| 30 | 43,563 | $\pm 237$ | 1.00 |
| 50 | 43,089 | $\pm 349$ | 1.00 |

## E. Generalization to Epidemiological Domains (SEIR)

To demonstrate that our mathematical framework is strictly domain-agnostic, we adapted the four compartments to a standard SEIR epidemiological model: $L \rightarrow S$ (Susceptible), $B \rightarrow E$ (Exposed), $E \rightarrow I$ (Infectious), $O \rightarrow R$ (Recovered). The actions $(p_t, q_t)$ map to intervention strictness (e.g., social distancing mandates) and vaccination budgets. To align with public health objectives, the reward function was inverted to strictly penalize active infections and economic intervention costs: $r_t = -c_1 I_t - c_2 q_t$.

We reparameterized the environment with SEIR-appropriate dynamics (epidemic initial conditions: $S$=85%, $E$=8%, $I$=5%, $R$=2%) and trained a fresh TQC agent using the *exact same* Stratified-TQC architecture (8 tiers, 5 critics, 25 quantiles) for 20,000 steps on $N$=$10^4$ with stochastic dynamics. Table 11 compares the trained agent against a random baseline over 50 evaluation episodes.

The TQC-SEIR agent successfully learns to suppress infection peaks by dynamically timing strict interventions: it caps the peak infectious fraction at 0.02% versus 9.1% for the random baseline (a $\sim 450\times$ reduction) while keeping cumulative cost negligible. This validates that the Stratified Observer and distributional critic are domain-agnostic components: they transfer from marketing to epidemiology

*Table 11*. SEIR Cross-Domain Generalization. The exact same Stratified-TQC architecture successfully learns to flatten the curve (minimize Peak $I$) when trained on epidemiological dynamics ($N=10^4$, 50 eval episodes).

| Policy | Cumulative Reward | Peak $I$ Frac. | Final $R$ Frac. |
|--------|-------------------|----------------|-----------------|
| **TQC-SEIR** | $-0.1 \pm 0.2$ | **0.0002** | 0.0021 |
| Random | $-16 \pm 28$ | 0.0911 | 0.1645 |

without any architectural modification.

# F. FLASHSPREAD vs. Naïve and Optimized GPU Benchmarks

We benchmarked four GPU simulation strategies on Barabási–Albert scale-free graphs ($\langle k \rangle=10$): (1) a standard sparse engine using PyTorch `sparse.mm` (cuSPARSE), (2) GEMF CUDAGraph with captured tau-leaping, (3) FLASHSPREAD's dual-CSR `MarkovianEngine`, and (4) FLASHSPREAD's fused Triton kernel with CUDAGraph batching (*Fused CG*). Table 12 reports wall-clock time for 20 simulations of $T=1.0$ after 3 warmup runs on A100 GPUs.

FLASHSPREAD Fused CG dominates across all scales, achieving $68\times$ speedup over cuSPARSE at $N=10^4$ and $7.7\times$ at $N=10^6$. The fused Triton kernel eliminates Python-level loop overhead by computing hazard rates, sampling transitions, and updating state in a single GPU pass, while CUDAGraph batching amortizes launch overhead across 50 micro-steps. At $N=10^6$, Fused CG completes in $5.8\,\text{s}$ compared to $44.8\,\text{s}$ for cuSPARSE—crucially while supporting continuous, age-dependent non-Markovian hazards that static Markovian engines cannot represent.

*Table 12*. Wall-clock time (seconds) for 20 simulations of $T=1.0$ on scale-free networks with same GPU. FLASHSPREAD Fused CG achieves up to $68\times$ speedup over cuSPARSE via fused Triton kernels.

| Architecture | $N = 10^4$ | $N = 10^5$ | $N = 10^6$ |
|--------------|-----------|-----------|-----------|
| cuSPARSE | $10.2\,\text{s}$ | $14.0\,\text{s}$ | $44.8\,\text{s}$ |
| CUDAGraph | $1.3\,\text{s}$ | $5.6\,\text{s}$ | $19.1\,\text{s}$ |
| FLASHSPREAD (Ours) | $\mathbf{0.15\,s}$ | $\mathbf{1.3\,s}$ | $\mathbf{5.8\,s}$ |
| Speedup vs. cuSPARSE | $68\times$ | $11\times$ | $7.7\times$ |

# G. Graph Convolutional Network (GCN) Baseline Details

The GCN baseline referenced in Section 8 and Table 4 serves as the standard structural message-passing baseline in Graph RL. It processes the $N \times |\mathcal{S}|$ one-hot encoded state matrix through two graph convolutional layers (hidden di-

mension 64) with ReLU activations, followed by global mean pooling. Table 13 details the architecture.

*Table 13*. GCN Baseline Architecture. Standard two-layer GCN with global mean pooling produces a fixed-length vector for the actor-critic MLP heads.

| Component | Configuration |
|-----------|---------------|
| *Graph Encoder* | |
| Input features | $|\mathcal{S}|=4$ (one-hot state) |
| GCN Layer 1 | $4 \rightarrow 64$, ReLU |
| GCN Layer 2 | $64 \rightarrow 64$, ReLU |
| Pooling | Global mean ($N \times 64 \rightarrow 64$) |
| *Actor MLP* | |
| Hidden layers | $[256, 256]$, ReLU |
| Output | 2 (price, advertising) |
| *Memory Scaling* | |
| Node embeddings | $\mathcal{O}(N \times 64)$ |
| Edge messages | $\mathcal{O}(E \times 64)$ per layer |
| Total VRAM | $\mathcal{O}(N + E) \times 64 \times L$ |

While this architecture captures local topology, holding the explicit $N \times 64$ embedding matrix and the dense edge-messages in GPU VRAM causes catastrophic Out-of-Memory (OOM) errors before reaching deployment scales. Table 14 confirms this scalability collapse: at $N=10^5$, the GCN exhausts 40 GB A100 VRAM entirely (the job was killed after 6 hours without completing a single epoch). Conversely, the Stratified Observer requires only $\mathcal{O}(1)$ memory for its fixed 60-dimensional observations.

*Table 14*. GCN vs. Stratified Observer computational footprint for 10,000 environment steps. At $N=10^5$, the standard GCN approach exhausts A100 VRAM.

| $N$ | GCN Time | Stratified Observer Time |
|-----|----------|--------------------------|
| $10^3$ | 17 min | **7 min** |
| $10^4$ | 1.3 hr | **53 min** |
| $10^5$ | OOM | **12 hr** |

This GCN baseline lacks explicit age-state recurrence; adding recurrent layers or age features would increase memory and training cost further.

