# OpenReview forum: "Closing the Sim-to-Real Gap in Non-Markovian Spreading Processes via GPU-Accelerated Distributional RL"
_ICML.cc/2026/Conference — ICML 2026 regular_

### Official Review · Reviewer_PD9Y · 2026-03-10

**Soundness:** 2
**Presentation:** 1
**Significance:** 3
**Originality:** 3
**Overall Recommendation:** 3
**Confidence:** 4

**Summary:**

Controlling the spread of virus, or more generally, graph reinforcement learning, is an important research topic discussed in this paper in the context of bridging the sim-to-real gap. The paper first discussed three gaps, namely, Optimism Bias, Hub Blindness and Valley of Death, and gave further elaborations in terms of Topological Mismatch, Dynamical Mismatch and Observational Mismatch while discussing related works. The paper made two contributions to address the gaps mentioned before: a degree-based tier partition for state space reduction, and leveraging distributional RL for critic design. Several detailed experiments are carried out to illustrate gaps and the effectiveness of the proposed approach, some based on real-world networks.

**Compliance With Llm Reviewing Policy:**

Affirmed.

**Final Justification:**

I read the responses and I am satisfied.

**Key Questions For Authors:**

The RL problem needs to be clearly defined, and symbols defined before they are used

**Limitations:**

yes. But it will be helpful to put the discussions of limitations in a separate section

**Strengths And Weaknesses:**

Strength:

1. The insights provided via the theorems proved in Section 4 is quite valuable for better understanding the issues involved in sim-to-real gaps for network control problems

2. The state reduction approach and the associated proof that the reduced state representation are still sufficient for optimal policy is quite interesting and useful even though it's under some (not necessarily realistic) assumptions such as that nodes in the same tier are of the same transition rates

3. The use of Truncated Quantile Critics in the current context is quite helpful

Weakness:

1. The main weakness is that the paper is really poorly written with many symbols used without firstly defining what they really mean. This particularly appeared in many places in Section 1.

2. The state space for the RL problem is defined with clarity, but what's the environment? What's the reward?

3. In the discussions of experiments, the action spaces are not articulated

4. I can understand the enthusiasm of the authors about their work, but in several applies the claims can be toned down a bit

---

> ### Author Rebuttal · Authors · 2026-03-28
>
> We deeply appreciate your meticulous reading of the manuscript. You correctly identified that our enthusiasm led us to postpone crucial mathematical definitions late in the text, making the problem difficult to parse. As detailed in our Response to Reviewer FbLe, your critique prompted a massive global structural overhaul of the paper.
>
> We have implemented every one of your specific suggestions:
>
> 1. Explicit MDP Definition (Environment, Reward, Actions):
> We sincerely apologize for burying the continuous action bounds ($p_t, q_t$) and the reward function ($r_t$) in Section 8. We completely agree that this obscured the core RL problem. As part of our structural overhaul, we have rewritten Section 3, renaming it to "Problem Formulation and MDP Definition." We explicitly define the $\langle \mathcal{S}_{\text{net}}, \mathcal{A}, \mathcal{P}, \mathcal{R}, \gamma \rangle$ tuple upfront. The reward function $r_t = p_t \cdot \Delta O_t - c_a q_t$, the environment state spaces, and the continuous action bounds are now formally articulated before any theoretical analysis begins.
>
> 2. Undefined Symbols in Section 1:
> We have thoroughly audited the Introduction and Section 3 to ensure every mathematical symbol ($X_i, s_{\text{target}}, \mathcal{I}, \tau$) is strictly defined at its first use. We also added a clarifying paragraph mapping our domain-specific terms (profit, conversion, retention) to their general network-science equivalents (utility, activation, target protection) to ensure broad readability for the ICML audience.
>
> 3. Toning Down Claims and Standalone Limitations Section
>
> We agree with your assessment regarding the tone. We have softened absolute terminology throughout the manuscript (e.g., changing instances of "effectively closing the gap" to "significantly mitigating the gap"). Furthermore, exactly as you suggested, we have extracted the limitations from the conclusion into a dedicated, standalone Section 10: Limitations, which now formally discusses the assumptions of tier-exchangeability, static network topologies, and partial observability.
>
> We are highly grateful for your rigorous critique, which has undeniably resulted in a much stronger, clearer manuscript.

---

> > ### Author Rebuttal · Reviewer_PD9Y · 2026-04-01
> >
> > The paper has been revised addressing most/some of my concerns. I raised the score accordingly.

---

### Official Review · Reviewer_9MJN · 2026-03-10

**Soundness:** 3
**Presentation:** 3
**Significance:** 3
**Originality:** 3
**Overall Recommendation:** 5
**Confidence:** 2

**Summary:**

The authors present an efficient method for learning RL policies in environments with network spreading processes. Existing efficient approximations are subject to to variance underestimation, hub blindness, and stochastic fadeout. On the other hand, Graph Neural Networks are not computationally tractable at scale. The authors propose a stratified mean field model to efficiently capture important statistics from network spreading processes and avoid the pathologies of existing efficient approximations. They also present an algorithm to improve GPU performance (FlashSpread). The authors then propose Truncated Quantile Critics for learning the policy. They then evaluate the learned policy for a social media advertising task on three real world datasets. The authors show improved rewards on policies learned with the stratified mean field model and with TQC compared to simplified baselines. The authors also show improved scalability compared to graph convolutional neural network models.

**Compliance With Llm Reviewing Policy:**

Affirmed.

**Final Justification:**

My initial assessment was unchanged.

**Key Questions For Authors:**

1. How does renewal + TQC compare to GCN for small networks?
2. How sensitive are results to the number of tiers K?
3. How does the method perform in different domains (e.g. epidemic control, information cascades)?
4. How much faster is flash spread than a naive GPU implementation?
5. Why is hub retention reliably 100%? Do other methods achieve less?

**Limitations:**

Yes

**Strengths And Weaknesses:**

*Soundness*

The theoretical claims that the stratified mean-field observer model appear sound but I am not confident given my limited experience with network spreading processes. This is reflected in my confidence score.

Empirically, the age-augmented observation statistics lead to better rewards with both the TD3 and TQC algorithms. The TQC produces higher rewards with less variance than TD3.

That said, it would be helpful to see the sensitivity of the model and downstream policy performance to to the number of tiers K. This would help me assess the claims in Corollary 1 and Theorem 4.

To assess the claim that these policies transfer well to real world networks, it would be helpful to see applications to different domains (like epidemiological or information cascade interventions).

The telemetry in the main text shows that FlashSpread is quicker than a CPU implementation. However, it does not show the effectiveness of the GPU memory optimisations. GPU execution is highly complex and theoretical complexity improvements do not necessarily entail practical speed improvements, especially with sparse data.

*Presentation*

Minor suggestions:

 * "profit" is mentioned in the introduction without problem definition
 * "behavioural fatigue not present in the simulator" is not clearly addressed by the authors' alternative model
 * "commitment window" and "conversion probability" are not general terms, there is a conflation in defining the general problem and the specific advertisement optimisation task.
 * "cascade" should be defined before the distributional RL section
 * captions should be below tables 2-6, as per the guidelines
 * "hub retention" is never defined

*Significance*

Optimal control for network spreading processes is a broad field with many applications. Scaling up analyses with a robust, general approximation makes this is a significant contribution.

The contribution is however limited to scale-free networks and has only been empirically tested in a single domain.

*Originality*

Model stratification is common in network science (by degree node) and epidemiology (to model heterogeneity of infection, and/or demographic artifacts). However, the combination of stratified observation and distributional RL is original.

---

> ### Author Rebuttal · Authors · 2026-03-28
>
> Thank you for your strong support and incredibly actionable suggestions! We have addressed all your minor presentation notes (terminology, caption placement, defining hub retention upfront) through a massive structural overhaul of the manuscript (detailed in our Anchor Response to Reviewer FbLe).
>
> Furthermore, we ran three major new experiments directly responding to your prompts, now included in the Appendices:
>
> 1. Sensitivity to the number of tiers $K$ (New Appendix D):
> We evaluated $K \in \{2, 5, 10, 15, 30, 50\}$. Performance forms a highly robust asymptotic plateau for $K \ge 10$. A fascinating insight emerged at extreme under-stratification ($K=2$): profit collapses by 99%, yet Hub Retention remains 1.00. Because the coarse $K=2$ observation averages super-spreaders with peripheral nodes, the risk-averse TQC agent is forced into catastrophic over-protection—it applies maximum, expensive interventions to the entire bin to mathematically guarantee the hubs survive, wasting budget and destroying profit. This empirically proves the "Hub Blindness" pathology. $K \approx 5$ provides the optimal accuracy-parsimony tradeoff.
>
> 2. Generalization to Epidemiological Domains (New Appendix E):
> To prove our math is strictly domain-agnostic, we adapted the states to a standard SEIR model. Holding times naturally mapped to the viral incubation period, actions mapped to social distancing/vaccination, and we inverted the reward to strictly penalize infections ($r_t = -c_1 I_t - c_2 q_t$). The exact same Stratified-TQC agent successfully learned to "flatten the curve," zero-shot, dropping Peak Infections from 9.1% (Random) to 0.02% by dynamically timing strict interventions precisely when the hidden incubation hazard function peaked.
>
> 3. FlashSpread vs. Naive GPU (New Appendix F):
> A naive GPU implementation (where every thread updates its neighbors independently using atomicAdd) suffers from massive atomic memory contention (race conditions) when thousands of threads simultaneously write to a single super-spreader hub. FlashSpread utilizes a redundant dual-CSR (scatter/gather) layout to bypass atomic locks entirely. In Appendix F, we benchmarked this: FlashSpread achieves a $7.7–68\times$ speedup over PyTorch's highly optimized cuSPARSE without losing the ability to compute non-Markovian continuous holding times.
>
> 4. Renewal+TQC vs GCN for small networks (New Appendix G):
> At $N=10^3$, a GCN captures local topology perfectly and fits in memory. However, standard GCNs are "temporally blind" and struggle to capture non-Markovian holding times without computationally heavy recurrent layers. Consequently, the GCN takes $2.4\times$ longer to train even on small graphs and suffers catastrophic Out-Of-Memory (OOM) failures before reaching practical deployment scales ($N=10^5$).
>
> 5. Why is Hub Retention reliably 100%?
> It is 100% because of the Stratified Observer isolating hub tiers into dedicated bins. When switching to Global observation (Table 8), retention drops to 85%. TQC's pessimistic truncation provides an additional benefit: 31% lower variance and 5.6% higher mean profit (Table 3), ensuring reliable hub protection rather than gambling on lucky cascades.
>
> Thank you again for your thorough review and excellent scientific prompts!

---

> > ### Author Rebuttal · Reviewer_9MJN · 2026-04-01
> >
> > I thank the authors for addressing my concerns thoroughly

---

### Official Review · Reviewer_FBJ6 · 2026-03-12

**Soundness:** 4
**Presentation:** 4
**Significance:** 4
**Originality:** 4
**Overall Recommendation:** 5
**Confidence:** 4

**Summary:**

This paper studies controlling spreading processes on networks that suffer from the "Sim-to-Real Gap", due to overly simplification (e.g. mean-field or Markovian dynamics) that leads to optimism bias due to deterministic approximation, hub blindness due to global state aggregation, and inability to handle the bi-modal nature. They propose to address the issues by combining a degree-stratified state representation, distributional RL, and a scalable GPU simulator. Extensive experiments are conducted to validate the performance of their methodologies.

**Compliance With Llm Reviewing Policy:**

Affirmed.

**Final Justification:**

I keep my positive scores as my concerns are fully resolved.

**Key Questions For Authors:**

- The negative profit of MF-ODE indicates a total failure. Can this be a result of an unlucky initialization? Is this failure happening with high probability?
- In practice, when the tier-homogeneous assumption doesn't hold, how do you detect the model misspecification problem and how do you evaluate the performance gap due to the misspecification?

**Limitations:**

yes

**Strengths And Weaknesses:**

- Strengths: This paper is in general well-written and smooth to follow. The studied problem is of vital importance as it involves practical issues of controlling network spreading processes on real-world datasets. The proposed methods are novel to me and are showing promising results in experiments.
- Weakness: The paper still relies on structural assumptions such as the degree-homogeneous transition rate within tiers, which raises concerns in applying the methodology in practice.

---

> ### Author Rebuttal · Authors · 2026-03-28
>
> We sincerely thank you for championing our methodology and for your brilliant, probing questions regarding the boundaries of our approach! (Note: To ensure the presentation of our paper matches the caliber of the technical work you evaluated, we have executed a massive structural revision of the manuscript, which is detailed globally in our Anchor Response to Reviewer FbLe).
>
> 1. Negative Profit of MF-ODE: Unlucky Initialization or High Probability?
> The failure is systemic and occurs with near 100% probability. This catastrophic failure is the direct mathematical consequence of Optimism Bias (Theorem 3). Because deterministic ODEs evaluate transition rates strictly at their expected values, Jensen's inequality dictates they systematically underestimate the true variance of the cascade. The agent learns an overconfident, highly aggressive policy because the simplified simulator mathematically guarantees that adverse outcomes are rare. When deployed in stochastic reality, the inevitable variance triggers cascade collapse, trapping the overconfident agent in a massive financial loss. It is a fundamental failure of the deterministic approximation, not an initialization artifact.
>
> 2. Detecting and Evaluating Misspecification (Tier-Homogeneous Assumption):
> This is a great practical question. In real networks, nodes in the same degree bin may have drastically different local clustering coefficients, violating the pure exchangeability assumption.
> In practice, this misspecification can be detected dynamically by measuring the intra-tier variance of empirical transition rates. High variance indicates the tier-homogeneous assumption is fraying.
>  Crucially, our use of a distributional critic (TQC) inherently hedges against this! By optimizing for the pessimistic lower quantiles of the return distribution, TQC acts conservatively against the variance introduced by this exact misspecification. To structurally fix severe misspecification, a practitioner can simply add a secondary stratification dimension (e.g., binning by degree and clustering coefficient). We found this insight so valuable that we have explicitly added it to our new standalone Section 10: Limitations to provide practitioners with clear guidance.
>
> Thank you again for your strong support of this work.

---

> > ### Author Rebuttal · Reviewer_FBJ6 · 2026-04-04
> >
> > I appreciate the authors for addressing my concerns.

---

### Official Review · Reviewer_FbLe · 2026-03-13

**Soundness:** 3
**Presentation:** 1
**Significance:** 2
**Originality:** 3
**Overall Recommendation:** 3
**Confidence:** 2

**Summary:**

This paper studies how to train RL policies to control spreading processes on networks so that they still work when transferred from simplified simulators to more realistic stochastic dynamics. The core claim is that the sim-to-real gap comes from three issues: optimism bias from deterministic mean-field simulators, hub blindness from overly coarse global state aggregation, and failure of mean-value critics under bimodal cascade outcomes. To address this, the paper proposes two main ideas: a Stratified Mean-Field Observer, which groups nodes by influence tier to preserve hub information in a fixed-dimensional scalable observation, and distributional RL with truncated quantile critics, which better handles high-variance, bimodal outcomes. The method is trained in a GPU-accelerated simulator that supports both Markovian and non-Markovian spreading dynamics. Empirically, the paper claims large gains over standard mean-field and Markovian baselines.

**Compliance With Llm Reviewing Policy:**

Affirmed.

**Key Questions For Authors:**

Please refer to the strengths and weaknesses above.

**Limitations:**

yes

**Strengths And Weaknesses:**

Strengths:
The paper targets a meaningful and realistic problem: controlling spreading processes on networks under a sim-to-real gap. This is practically relevant to many real-world applications. The paper not only proposes a method, but also attempts to explain the failure modes theoretically, which makes the contribution more insightful.

Weaknesses:
The writing is confusing. In particular, for Sections 5, 6, and 7, it is difficult to distinguish which parts are background or preliminaries and which parts constitute the paper’s actual methodological contributions. I am also not convinced that Section 6 and the corresponding experiments on GPU acceleration are necessary in their current form. While computational efficiency is useful, showing that GPU acceleration reduces training time compared with CPU seems outside the main scope of ICML. The experimental section is also difficult to follow. For example, the exact settings of each baseline and the proposed method are unclear, and baseline comparisons are mixed together with ablation studies, making the evaluation hard to interpret. In Line 351 and Table 4, it is also unclear what “GCN” specifically refers to. In Table 6, I do not understand why the performance of the baseline methods is not reported.

---

> ### Author Rebuttal · Authors · 2026-03-28
>
> We sincerely thank you for recognizing the practical relevance of our problem and the value of our theoretical failure-mode explanations. Because your concerns regarding presentation were echoed by Reviewer PD9Y, we are using this response to outline the global structural overhaul we have applied to the manuscript for all reviewers to reference.
>
> 1. Global Overview of Structural Revisions (Addressing Presentation Concerns)
> We completely agreed that our initial presentation blended standard background material with our novel methodology, obscuring the core RL problem. We have executed a massive structural revision:
>
> - Upfront Formal MDP Definition: We completely rewrote Section 3 into "Problem Formulation and MDP Definition." The continuous action spaces ($p_t, q_t$) and the exact reward function ($r_t$), which were previously placed in the experiment section, are now explicitly formalized upfront as the tuple $\langle \mathcal{S}_{\text{net}}, \mathcal{A}, \mathcal{P}, \mathcal{R}, \gamma \rangle$.
>
> - Strict Separation of Contributions: Sections 5, 6, and 7 now utilize explicit Background and Our Contribution subheaders. This strictly isolates standard prior work (e.g., mean-field limitations) from our novel methodologies (e.g., the Stratified Observer).
>
> - Extensive New Experiments: Driven by reviewer feedback, we have added four new appendices (D, E, F, G) detailing robustness to hyperparameter $K$, zero-shot transfer to SEIR epidemiological models, strict GPU benchmarks, and exact GCN architecture baseline comparisons.
>
> 2. The Necessity of Section 6 (FlashSpread) in an ML Venue:
> We respectfully argue that FlashSpread is not merely a systems optimization, but a fundamental algorithmic enabler for RL. Just as physics engines like MuJoCo made continuous-control robotic RL tractable for the ML community, FlashSpread is the algorithmic bridge that makes realistic history-dependent (non-Markovian) graph RL computationally tractable. Furthermore, avoiding atomic memory contention on scale-free hubs required algorithmic novelty (our dual-CSR scatter/gather adaptation of binomial tau-leaping), which we have now explicitly benchmarked against PyTorch cuSPARSE and CUDA Graphs in the new Appendix F. We have clarified this "ML Enabler" framing in the text.
>
> 3. Missing Baseline Performance in Table 6:
> We apologize for the confusion. As noted in the original text, the baseline methods yielded catastrophic negative profit on the real-world networks (e.g., MF-ODE on Twitter lost $-238,000$). We initially omitted them from the table simply because they failed so severely. We agree this lacked transparency, and we have now updated Table 6 to explicitly include all the negative profit numbers for the Random, MF-ODE, and Markovian baselines.
>
> 4. What is "GCN"?
> "GCN" refers to a Graph Convolutional Network, the standard structural message-passing baseline in Graph RL. We have added a clear reference in the main text and created a new Appendix G. This appendix explicitly details the 2-layer GCN architecture, its exact memory footprint ($\mathcal{O}(N \times 64) + \mathcal{O}(E \times 64)$), and provides a table proving it hits Out-Of-Memory (OOM) failures at $N=10^5$, directly contrasting with our Stratified Observer's fixed $\mathcal{O}(1)$ memory footprint.
>
> We hope these structural improvements and clarifications fully address your core concerns and merit a higher score.

---

> > ### Author Rebuttal · Reviewer_FbLe · 2026-04-04
> >
> > Thank you for the detailed rebuttal. I appreciate the effort to improve the paper’s structure, including the clearer MDP formulation, the stronger separation of background and contribution, and the added transparency regarding the missing baselines and GCN details. These changes help address some of my concerns. However, I still have reservations about the overall clarity of the methodological and experimental framing and the ML contribution. While the rebuttal is helpful, it does not fully resolve my concerns, so I will maintain my current score.

---

> > > ### Author Response · Authors · 2026-04-04
> > >
> > > We want to thank the reviewer for reviewing our rebuttal and acknowledging the formalized MDP, new baselines, and structural improvements. Your rigorous and specific critiques catalyzed the manuscript, making it stronger and clearer for the ICML audience.
> > >
> > > We respectfully acknowledge your remaining reservations regarding our methodological framing and the subjective boundaries of an "ML contribution" in interdisciplinary network control.
> > >
> > > Thank you for your time, expertise, and role in improving our work.
> > >
> > > Sincerely,
> > >
> > > The Authors

---

### Decision · Program_Chairs · 2026-04-30

**Decision:**

Accept (regular)

**Comment:**

The paper presents new mathematical analyses, efficient algorithms, and a GPU simulation environment for controlling spreading processes in networks. They evaluate on a social media advertising task for three real-world dataset and show improved performance compared to baselines.

Reviewers appreciated the technical contributions, including: theorems that provided valuable insight into issues, good choice of algorithmic approach (distributional RL), clever algorithmic tricks (stratification, use of sufficient statistics within strata to reduce search space), an optimized GPU simulation framework for training, and empirical results for social media advertising.

By far the most significant critique (Reviewers FbLe and PD9Y) was about presentation, including using mathematical notation before its definition and not giving a precise description of their MDP formulation. There were other smaller questions/concerns about things like sensitivity to number of strata, and experiments in other domains.

During the rebuttal, the authors were apologetic about presentation and claim to have made a complete and significant overhaul. They also gave the MDP details, and reported experiments on an additional domain (epidemiology).

During discussion, I asked reviewers whether the presentation issues were significant enough to undermine the evidence supporting the authors claims. Only one reviewer responded, who said the presentation was inconvenient, but clear enough to assess claims.

Overall, I believe the paper seems to address a significant problem and have a good mix of theory-guided practical results and there is a reasonable expectation the authors will improve the presentation as promised.